# Artificially stimulating retrotransposon activity increases mortality and accelerates a subset of aging phenotypes in *Drosophila*

Joyce Rigal, Ane Martin Anduaga, Elena Bitman, Emma Rivellese, Sebastian Kadener, Michael T Marr*

Biology Department, Brandeis University, Waltham, United States

**Abstract** Transposable elements (TEs) are mobile sequences of DNA that can become transcriptionally active as an animal ages. Whether TE activity is simply a by-product of heterochromatin breakdown or can contribute toward the aging process is not known. Here, we place the TE *gypsy* under the control of the UAS GAL4 system to model TE activation during aging. We find that increased TE activity shortens the life span of male *Drosophila melanogaster*. The effect is only apparent in middle-aged animals. The increase in mortality is not seen in young animals. An intact reverse transcriptase is necessary for the decrease in life span, implicating a DNA-mediated process in the effect. The decline in life span in the active *gypsy* flies is accompanied by the acceleration of a subset of aging phenotypes. TE activity increases sensitivity to oxidative stress and promotes a decline in circadian rhythmicity. The overexpression of the Forkhead-box O family (FOXO) stress response transcription factor can partially rescue the detrimental effects of increased TE activity on life span. Our results provide evidence that active TEs can behave as effectors in the aging process and suggest a potential novel role for dFOXO in its promotion of longevity in *D. melanogaster*.

*For correspondence:
mmarr@brandeis.edu

**Competing interest:** The authors declare that no competing interests exist.

## Editor's evaluation

The work presented in this article clearly shows that the *Drosophila* FOXO gene when expressed can mitigate the effects of enhanced transposon activation on aging. Using the transposon gypsy under the expression of the UAS Gal4 the work shows convincingly that such expression decreases life span. The effects of such transposition are dependent upon reverse transcription and by an unknown pathway the FOXO can mediate the aging pathway(s) as the later protein seems not to be controlling transposon expression.

## Introduction

Aging leads to a progressive loss of physiological integrity that culminates in a decline of function and an increased risk of death (*López-Otín et al., 2013*). It is a universal process that involves the multifactorial interaction of diverse mechanisms that are not yet fully elucidated. Transposable elements (TEs) are among the many factors that have been proposed to be involved in aging (*Morley, 1995*; *Wood and Helfand, 2013*). TEs are present in every eukaryotic genome sequenced to date (*García Guerreiro, 2012*; *Huang et al., 2012*). They are sequences of DNA that can move from one place to another (*McCLINTOCK, 1950*), either by reverse transcription and insertion into the genome (Class 1: retrotransposons), or through direct excision and movement of the element (Class 2: DNA TE) (*McCullers and Steiniger, 2017*).

Multiple studies (*Gorbunova et al., 2021*) report that TE mRNA levels increase in the aging somatic tissue of flies (*Giordani et al., 2021*; *Li et al., 2013*; *Wood et al., 2016*), termites (*Elsner et al., 2018*), mice (*De Cecco et al., 2013b*), rats (*Mumford et al., 2019*), and humans (*LaRocca et al., 2020*). A direct correlation to an increase in genomic copy number has been difficult to determine (*Treiber and Waddell, 2017*; *Yang et al., 2022*). Nonetheless, clear evidence for an increase in TE somatic insertions with age has been obtained using reporter systems. Two reporter systems for insertions of the long-terminal-repeat (LTR) retrotransposon *gypsy* demonstrate that *gypsy* insertions increase during aging in the *Drosophila melanogaster* brain and fat body (*Chang et al., 2019a*; *Li et al., 2013*; *Wood et al., 2016*).

TE movement in somatic tissue has been proposed to be a driver of genomic instability through DNA damage (*Ivics and Izsvák, 2010*) and potentially aging (*Morley, 1995*; *Wood and Helfand, 2013*; *Woodruff and Nikitin, 1995*). Additionally, TE activity has been reported to cause disease in humans (*Hancks and Kazazian, 2012*). Current research reports that long-interspersed-element-1 (*L1*) activity itself, without an increase in insertions, can trigger an inflammation response that contributes to aging-related phenotypes in human senescent cells (*De Cecco et al., 2019*). In aged mice, the use of reverse transcriptase inhibitors can downregulate this age-associated inflammation (*De Cecco et al., 2019*) implicating retrotransposons. The shortened life span of a Dicer-2 (*dcr-2*) mutant fly strain, which has an increase in TE expression, can also be extended by the use of reverse transcriptase inhibitors (*Wood et al., 2016*). In summary, current evidence suggests a role for TE activity in aging.

However, whether the role of TE activity is as effector or bystander of the aging process is an open question. As an animal ages, heterochromatin-repressive marks decrease and resident silent genes can become expressed (*Jiang et al., 2013*). TE sequences are enriched in silent heterochromatin and thus become expressed (*De Cecco et al., 2013a*). This raises the question of whether TE expression is simply a by-product of age-related heterochromatin breakdown or whether TE themselves can contribute to the aging process. To date, this has not been directly assayed.

To combat the effects of TE activity, cells have evolved small RNA pathways to maintain silencing of TE. The PIWI pathway dominates in the germline while the somatic tissue of *Drosophila* is thought to mainly rely on the siRNA pathway (*Hyun, 2017*). This pathway is based on Dicer-2 cleaving double-stranded (ds) RNA precursors, generally viral genomes or TE dsRNA, into small RNAs that are loaded into Ago2 guiding the RNA-induced silencing complex (RISC) to cleave its targets (*Hyun, 2017*). In *D. melanogaster*, endogenous siRNAs in RISC mapping to TE loci have been reported (*Czech et al., 2008*; *Ghildiyal et al., 2008*; *Kawamura et al., 2008*). The mutation, knockdown, or depletion of genes involved in the siRNA pathway such as *Dcr-2* (*Lee et al., 2004*) and *AGO2* (*Okamura et al., 2004*) leads to increased expression of TE in somatic cells and a shortened life span (*Kawamura et al., 2008*; *Czech et al., 2008*; *Chung et al., 2008*; *Lim et al., 2011*; *Li et al., 2013*; *Wood et al., 2016*; *Chen et al., 2016*). On the other hand, the overexpression of *Dcr-2* (*Wood et al., 2016*) and *AGO2* (*Yang et al., 2022*) in adult fly somatic tissue can lower TE expression and extend life span. Taken together, these results suggest that TE activity can influence life span. The effect of TE activity on life span has not been directly determined.

The Forkhead-box O family (FOXO) is an evolutionarily conserved transcription factor capable of enhancing longevity by enabling the cell to respond to diverse stress signals (*Calnan and Brunet, 2008*). The longevity effects of FOXO have been reported in worms, flies, and mice (*Martins et al., 2016*). FOXO activity can enable transcriptional responses to provide protective effects against cellular stress: oxidative stress, heat shock, virus infection, and defects in protein homeostasis, among many others (*Donovan and Marr, 2016*; *Martins et al., 2016*; *Spellberg and Marr, 2015*). Importantly, dFOXO is an upstream activator of the small RNA pathways that limit TE expression in somatic tissues (*Spellberg and Marr, 2015*). Whether TE activity is a condition that FOXO activity might protect against is unknown.

In this study, we have set up a system to directly assay if TE can become active contributors of the aging process. For this, we placed the retrotransposon *gypsy* (*Bayev et al., 1984*), which will be used as a candidate to model TE activity during aging, under the control of the UAS GAL4 system (*Brand and Perrimon, 1993*). The retrotransposon *gypsy* is a TE that has been clearly shown to become active in aged *D. melanogaster* somatic tissue under natural conditions (*Li et al., 2013*; *Wood et al., 2016*). In our system, we find that active TE expression significantly increases the mortality in middle-aged

flies and that an intact reverse transcriptase is necessary for this effect. The increase in mortality is accompanied by acceleration of a subset of aging-related phenotypes. We find that the FOXO homologue in *Drosophila* (dFOXO) can counteract and partially rescue the decrease in life span generated by an active TE, suggesting a possible novel mechanism through which dFOXO might promote longevity in *D. melanogaster*.

## Results

To begin to investigate dFOXO's possible involvement in TE regulation, we sequenced the RNA from whole animals aged 5–6 days and 30–31 days, in $w^{DAH}$ (wt) and dFOXO deletion ($w^{DAH\ \Delta94}$) males (*Slack et al., 2011*). These lines have been extensively backcrossed, making them isogenic other than for the dFOXO deletion (*Slack et al., 2011*). We mapped the reads to the *Drosophila* genome and to a custom genome file that contains the consensus sequences for *Drosophila* transposons. Differentially expressed RNAs were identified using DESeq2. In these conditions, the wt animals display significantly

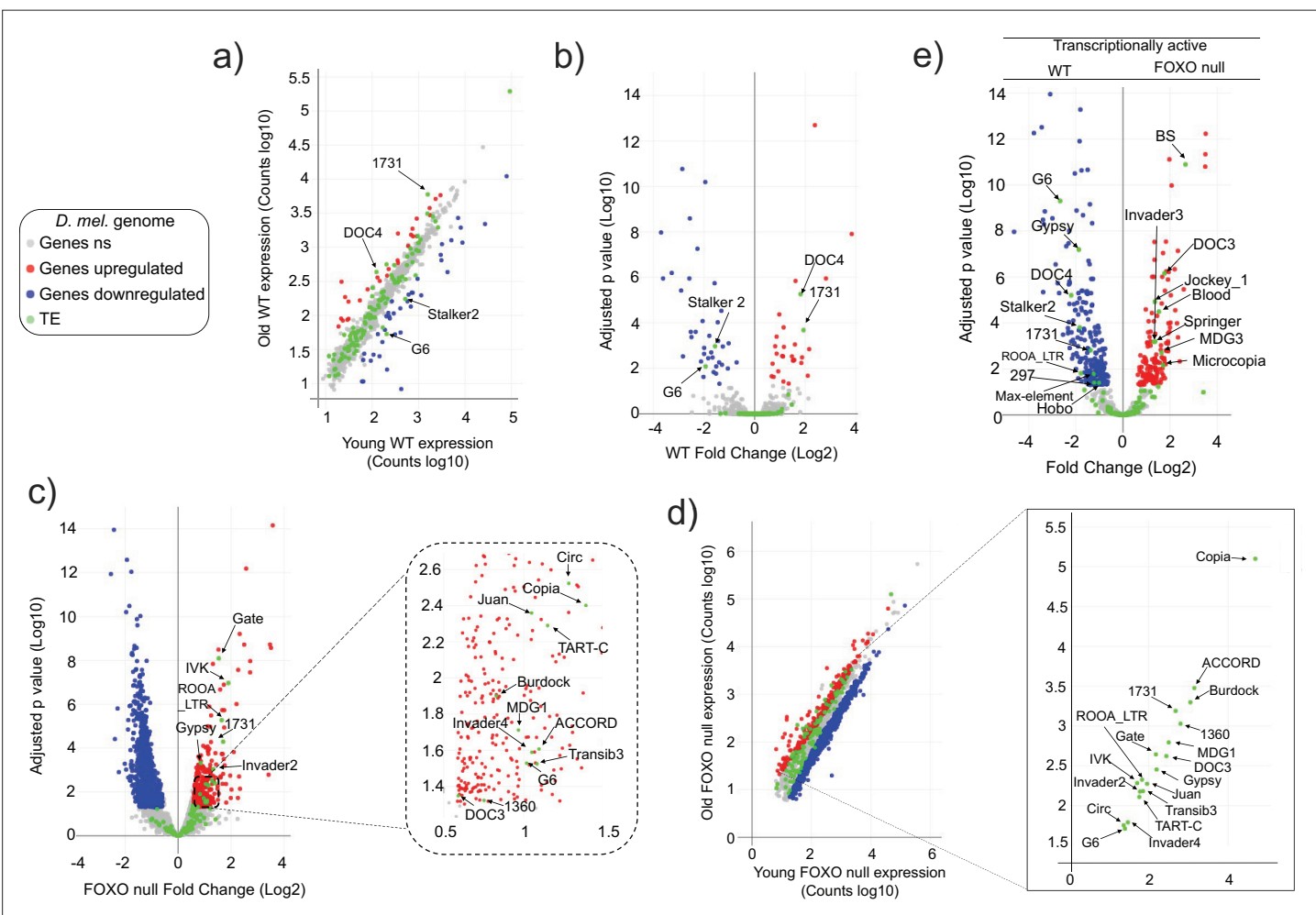

**Figure 1.** Transposable element (TE) expression increases with age in Forkhead-box O family (FOXO) null flies. Data represents RNAseq from three biological replicates. Legend (gray: fly genes not significant [ns]; red: upregulated fly genes; blue: downregulated fly genes; green: TE). Differential expression indicates a 1.5-fold change or higher and an adjusted p-value<0.05, as determined by DESeq2. (**a–d**) Red indicates upregulation with age. Blue denotes downregulation with age. Significantly different TEs are identified by name. (**a, b**) Wildtype (WT) control. Four TEs are labeled. (**c, d**) FOXO null. Eighteen TEs are labeled. (**e**) Volcano plot of young WT control and young FOXO null. Red and blue dots indicate gene relative expression in FOXO null compared to WT, respectively. TEs are marked by green dots.

The online version of this article includes the following figure supplement(s) for figure 1:

**Figure supplement 1.** Differentially expressed transposable element (TE) in wildtype.

**Figure supplement 2.** Differentially expressed transposable element (TE) in Forkhead-box O family (FOXO) deletion flies.

increased expression of only two TEs (the *1731* and *Doc4 retrotransposons*). At the same time, there is decreased expression of two TEs (the *Stalker2* and *G6 retrotransposons*) with age (*Figure 1a and b*, *Figure 1—figure supplement 1*). By contrast, in the dFOXO deletion animals, 18 TEs (the retrotransposons: *Accord, Circ, Gate, Copia, Gypsy, MGD1, RooA, 1731, Burdock, TartC, Invader2, Invader4, Juan, G6, IVK, Doc3;* and the *DNA transposons: TransIB3, 1360*) exhibited a significant increase in expression with age (*Figure 1c and d*, *Figure 1—figure supplement 2*) while no TE expression decreased with age. This indicates that the overall TE load is greater in the aged dFOXO deletion flies.

The vast majority of TE expression levels in both strains fall within the observed range of average gene expression (*Figure 1a and d*). Total TE expression undergoes a small change overall in both genotypes. Expression increased only 1.2-fold with age (*Supplementary file 1*) in wt flies and 1.41-fold in dFOXO null flies (*Supplementary file 2*). Of the two TEs that increased with age in wt flies, only *1731* exhibits an increase in dFOXO null flies. Among the two TEs that decreased with age in wt, *G6* showed the opposite effect on expression in dFOXO null flies while *Stalker2* showed no change with age in dFOXO null flies. The direct comparison of individual TE expression levels in young wt and dFOXO null flies indicates that despite being backcrossed (*Slack et al., 2011*) different TEs are being expressed in each strain (*Figure 1e*). This means that the otherwise isogenic lines have a different transcriptionally active TE landscape. Therefore, beyond the comparison of the number of transcriptionally active TEs, a direct comparison between wt and dFOXO null flies to determine the effect of dFOXO on any specific TE expression during aging is challenging.

Our difficulties to determine the effect of dFOXO on any specific TE highlights the need for a controlled system to test TE expression and regulation during aging. The *gypsy* retrotransposon was selected as a model system to study TE activity, and we developed a UAS-*gypsy* system. The structure of the ectopic UAS-*gypsy* system is depicted in *Figure 2a*. A direct comparison to the endogenous *gypsy* element can be observed in a simplified overview of three broad stages in the *gypsy* life cycle (parental element, propagation through transcription and reverse transcription, and new insertion). The presence of the UAS promoter in the ectopic *gypsy* allows the control of *gypsy* expression by mating to a GAL4-expressing strain. In addition, we inserted a unique sequence tag in the 3′ LTR of the ectopic *gypsy* to differentiate it from the endogenous copy of *gypsy* (*Bayev et al., 1984*). We inserted the UAS-*gypsy* construct in the VK37 attP site on chromosome 2 using the PhiC31 system (*Venken et al., 2006*). Using this approach, we found that ectopic *gypsy* expression is significantly induced when the line is crossed to a strain in which GAL4 is ubiquitously expressed under the *ubiquitin* promoter (*Ubi>gypsy*) (*Figure 2b*).

This tag also allows us to take advantage of the strand transfer that occurs during reverse transcription of the retroelement before integrating into a new site in the genome. This process replaces the UAS sequence with a 5′ LTR and transfers the sequence tag to the new LTR of the newly integrated *gypsy* element (*Weaver, 2008*; *Figure 2a*). Using this approach, we found that new insertions are created and can be specifically detected (*Figure 2c*) when UAS-*gypsy* is expressed (*Ubi>gypsy* genotype). The junction between the recombinant *gypsy* 5′ LTR and the *gypsy* provirus sequence is only formed when a complete insertion is made. Sanger sequencing of the qPCR insertion products demonstrates the specific detection of this junction, indicating that the UAS-*gypsy* element can transpose and is an active TE element (*Figure 2d*).

To characterize the spectrum of insertion sites of the UAS-*gypsy* construct and rule out any potential 'hotspot' effect, we took a targeted sequencing approach. Using a biotinylated primer hybridizing with the unique sequence tag oriented to read out from the newly integrated 5′ LTR (*Table 1*), we created Illumina next-generation sequencing libraries to sequence the genomic junctions (*Figure 3—figure supplement 1*). Libraries were prepared from three biological replicates of ten 14-day-old male flies each. Sequencing reads were sorted using the LTR sequence as a barcode. After removing the LTR sequence, the exact site of insertion was mapped back to the *Drosophila* reference genome. More than 11,000 insertion sites were identified. Sites were identified in all chromosomes with the fraction of insertions roughly correlating with the size of the chromosome (*Figure 3a*). Comparing the insertion sites with the genome annotation allowed us to classify the insertion sites. Also, 66% of the mapped insertion sites are in transcribed regions (*Figure 3b*). Of those sites, the majority are in intronic regions (*Figure 3c*). Because the data provides nucleotide resolution, we could also determine the six-nucleotide target site duplication that occurs at the junction between the *gypsy* LTR and the genome upon insertion (*Dej et al., 1998*). The target site duplication consensus determined from

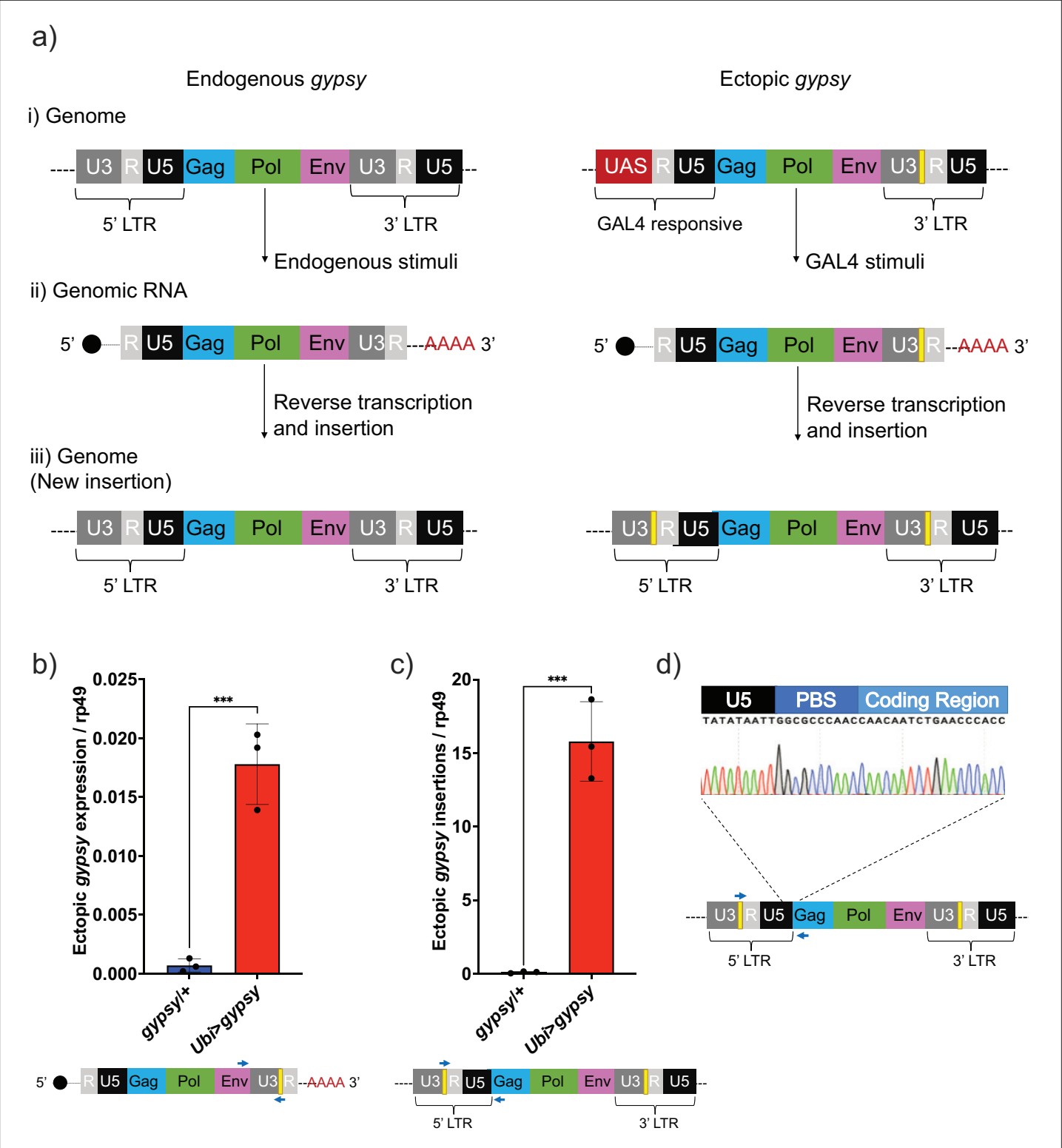

**Figure 2.** UAS-*gypsy* structure and functional test. (**a**) Simplified overview of three stages of the *gypsy* retrotransposon life cycle, (i) parent insertion in the genome, (ii) the transcribed RNA (genomic RNA), and (iii) the new copy inserted in the genome. The difference between the wildtype *gypsy* and ectopic *gypsy* resides in both its 5' and 3' long-terminal-repeat (LTR). The presence of an upstream activating sequence (UAS) in the 5' LTR allows *gypsy* to be transcribed in response to a GAL4 stimuli. In the 3' LTR, the addition of a unique sequence of DNA (denoted by a yellow square) not found in the *D. melanogaster* genome allows quantification and tracking of new insertions by molecular methods. (**b**) RT-qPCR of 5-day-old males. 3' end of ectopic

*Figure 2 continued*

*gypsy* transcript is detected. Data are represented as means ± SD (three biological replicates, each dot is a pool of five flies). One-tailed *t*-test, ***p-value=0.0005. (**c**) gDNA qPCR of 5-day-old males. Ectopic *gypsy* provirus insertion junctions are detected. Data are represented as means ± SD (three biological replicates, each dot is a pool of 10 flies). One-tailed *t*-test, ***p-value=0.0003. (**d**) Sanger sequencing of the newly created ectopic *gypsy* provirus insertion junction in *Ubi>gypsy* flies.

our mapped insertion sites matches the known YRYRYR previously identified for *gypsy* integration sites (*Figure 3d*; *Dej et al., 1998*). In addition, the alternating purine-pyrimidine sequence and AT-rich nature of the target sites are consistent with bent DNA, a hallmark of DNA transposons, retroviral integrases, and other transposases.

To determine the distribution of the insertion sites, we divided the largest chromosome arms (3R, 3L, 2R, 2L, X) into roughly 1 megabase bins and counted the number of insertions per bin. Then we plotted the fraction of the total insertions on that chromosome arm in each bin. A plot for the arms of chromosome 3 is shown in *Figure 3e*. The plots of the other chromosomes are shown in *Figure 3—figure supplement 2*. At this level of resolution, insertions are detected roughly evenly across the chromosomes.

The feasibility of the system allowed us to test whether TE activity could affect life span. We set up three independent cohorts to measure life span. The assays were performed at different times of the year. We consistently found that the somatic expression of an active *gypsy* significantly decreased the life span of male flies (*Figure 4a*). The individual cohorts showed a consistent effect on life span and the merged data was used to analyze the effect. A 19% reduction of life span in the active *gypsy* male flies is observed compared to parental controls, with a median survival of 70 days and 86 days, respectively. A life span effect was also observed in females; surprisingly it was also present in the UAS-*gypsy* parental control (*Figure 4—figure supplement 1*). Interestingly, a molecular assay demonstrates that more detected insertions are present in female UAS-*gypsy* controls than their male equivalents (*Figure 4—figure supplement 2*). An additional molecular assay was done in the *Ubi>gypsy* genotype to look at the distribution of detected insertions between the head and body (*Figure 4—figure supplement 3*). No significant difference was detected in male flies, indicating an equivalent level of insertions throughout somatic tissue. In females, however, a significant bias toward insertions in the body was detected, indicating possible differential expression of UAS-*gypsy* in the mixed tissues. Due to the confounding effects of the system observed in female flies (life span defect in parental UAS-*gypsy* control and differential distribution of detected insertions in *Ubi>gypsy*), male flies were used for all following experiments. The life span effect on the *Ubi>gypsy* flies only becomes apparent once the flies start to age. At 26 days old, the survival curves noticeably separate from the parental controls and calculation of the age-specific mortality shows a significant increase in mortality present in 26–75-day-old animals (*Figure 4b–e*).

Ectopic *gypsy* DNA was detected at different ages in the survival curve (5, 14, 30, 50, and 70 days old) with three different primer pairs. In *Figure 5a*, the recombinant 5′ LTR *gypsy* junction is targeted to detect complete ectopic *gypsy* elements. In *Figure 5b*, the 3′ fragment of ectopic *gypsy* is targeted for detection. The 3′ fragment comprises the region between the *env* gene and the tag in the 3′ LTR. Complete and incomplete *gypsy* elements are quantified by this approach. In *Figure 5c*, the wildtype *gypsy* env region is targeted. This approach captures both the endogenous and ectopic

**Table 1.** Primers for next-generation sequencing.

| | |
|---|---|
| MB2640 | Biotin-GTGAGGGTTAATTCTGAGCTTGGC |
| MB1192 | Phos-AGATCGGAAGAGCACACGTCTGA-3′ amino blocked |
| MB1019 | TCAGACGTGTGCTCTTCCGATCT |
| MB2669 | CCTACACGACGCTCTTCCGATCTNNNTTCTTCGCGTGGAGCGTTGA |
| MB583 | AATGATACGGCGACCACCGAGATCTACACTCTTTCCCTACACGACGCTCTTCCGATCT |
| MB2673 | CAAGCAGAAGACGGCATACGAGAT<u>TCCGAAAC</u>GTGACTGGAGTTCAGACGTGTGCTCTTCCGATCT |
| MB2674 | CAAGCAGAAGACGGCATACGAGAT<u>TACGTACG</u>GTGACTGGAGTTCAGACGTGTGCTCTTCCGATCT |
| MB2675 | CAAGCAGAAGACGGCATACGAGAT<u>ATCCACTC</u>GTGACTGGAGTTCAGACGTGTGCTCTTCCGATCT |

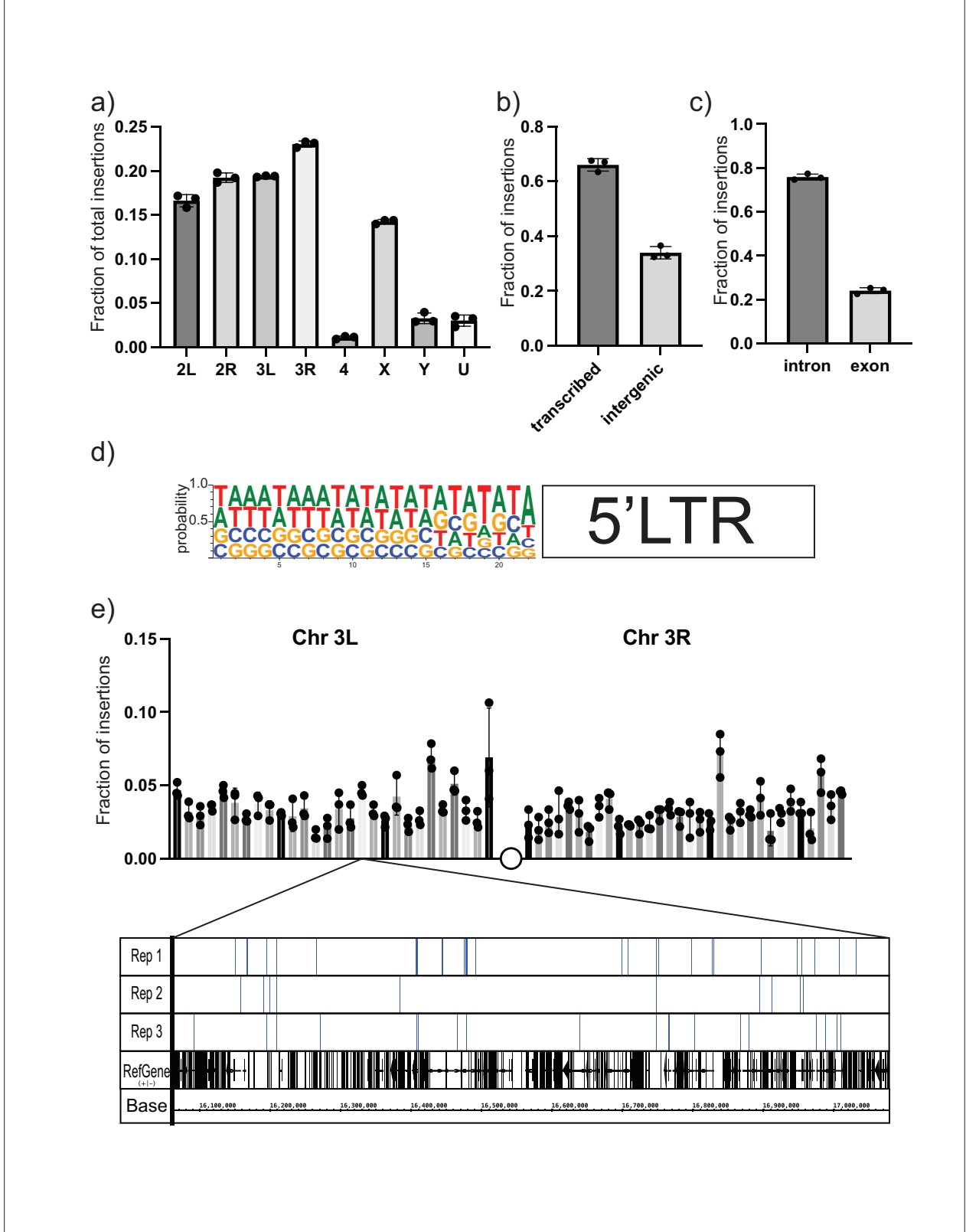

**Figure 3.** Next-generation sequencing (NGS) mapping of ectopic *gypsy* insertions. (**a**) The average fraction of total insertions is shown for each chromosome (4, X, Y) or chromosome arm (2L, 2R, 3L, 3R). In addition, the fraction mapping to unplaced contigs is indicated as U. (**b**) The fraction of total reads mapping to transcribed regions of the genome and intergenic regions is graphed. (**c**) The fraction of insertions that map to the transcribed regions of the genome is subdivided. Insertions mapping to regions annotated as introns or exons is graphed. (**d**) The sequences of the junction of the

*Figure 3 continued on next page*

*Figure 3 continued*

new 5′ long-terminal-repeat (LTR) and the *Drosophila* genome were aligned and used to determine the probability of finding each base at each position. These probabilities are indicated by the size of the letter at each position. (**e**) The fraction of insertions that map to each 1 megabase region of the reference genome for the arms of chromosome 3 is plotted. For illustration, a genome browser view of the 1 Mb in Chr3L is shown. The insertion sites in that region for each replicate are indicated. A collapsed track showing genes in that region is also shown. For all histograms, the bars represent the average of three biological replicates. Error bars indicate the standard deviation, and the filled circles indicate the individual measurements.

The online version of this article includes the following figure supplement(s) for figure 3:

**Figure supplement 1.** Schematic of targeted sequencing approach to map 5′ junctions.

**Figure supplement 2.** The fraction of insertions that map to each 1 megabase region of the reference genome for the arms of chromosome 2 and the X chromosome is plotted.

*gypsy* content in the genome. Of the three approaches, the detection of the 5′ LTR *gypsy* junction (*Figure 5a*) is the most stringent because only complete *gypsy* elements are detected. This is reflected in the smaller content of ectopic *gypsy* DNA detected compared to the 3′ fragment assay that detects both complete and incomplete *gypsy* elements (*Figure 5b and c*). The three approaches detect a relatively constant level of DNA while the animals are young with a surprising decrease in detection of ectopic *gypsy* DNA in older animals. Interestingly, ectopic *gypsy* RNA expression remains constant throughout the assayed time points (*Figure 5d*). Although variability increases greatly with age.

The effect on life span caused by TE activity could be due to the process of retrotransposition through a DNA intermediate or by disruption of RNA homeostasis. One way to address this is to remove the DNA synthesis step and test whether TE RNA presence alone can mediate the decrease in life span. This led us to test whether the life span effect would be present after deleting the reverse transcriptase (RT) from the UAS-*gypsy* polyprotein. We created a UAS-*gypsy* construct that contains an in-frame deletion in the polyprotein that removes most of the RT (*Marlor et al., 1986*) and inserted it in the same attP landing site on chromosome 2 (UAS-*ΔRT*). The deletion of RT in the UAS-*gypsy* transgene prevents the life span effect observed when crossed to *Ubi*-GAL4 (*Ubi>ΔRT* genotype) (*Figure 6a*). This is not due to a defect in mRNA levels because RT-qPCR indicates that *Ubi>ΔRT* and *Ubi>gypsy* express the transposon RNA at the same level (*Figure 6b*). Consistent with the loss of RT activity, we detect no increase in ectopic *gypsy* 3′ DNA in the *Ubi>ΔRT* line despite expressing equivalent levels of mRNA as *Ubi>gypsy* (*Figure 6c*). This result indicates that an active RT is required for the observed decrease in life span seen when *gypsy* is ectopically expressed in somatic tissue.

To determine whether the decrease in life span of the active *gypsy* flies was accompanied by an acceleration of aging-associated phenotypes, we decided to test whether the ectopic expression of the *gypsy* TE resulted in the early emergence of any aging hallmarks. In particular, we focused on four different phenotypes that appear in normal aging flies (decreased resistance to dietary paraquat; *Arking et al., 1991*), decline in negative geotaxis (*Rhodenizer et al., 2008*), decrease of total activity (*Sun et al., 2013*), and disruption of circadian phenotypes (*Curran et al., 2019*; *Rakshit et al., 2012*).

For the paraquat resistance assays, 5-, 15-, 30-, and 50-day-old flies were exposed to 20 mM paraquat. As expected, resistance to paraquat exposure decreased with age in all genotypes (*Figure 7a*). Interestingly, the survival curve of the 5-day-old active *gypsy* flies mimicked that of the 15-day-old parental control curves. Meanwhile, the 15-day-old active *gypsy* flies had a significant decrease in resistance to paraquat exposure when compared to age-matched parental controls. However, once the flies are 30 and 50 days old it appears TE activity can no longer exacerbate the oxidative stress survival as experimental and control flies die at a similar rate.

We also tested total activity as well as other well-characterized circadian phenotypes such as their ability to anticipate day and night, strength of their free-running conditions, or periodicity. Briefly, we placed 20-, 30-, or 40-day-old flies into *Drosophila* Activity Monitors and recorded their total activity for over 10 days at 25°C in 12:12 light–dark (LD) conditions. We did not find any significant decrease in the overall activity levels of the active *gypsy* flies compared to their parental controls at any of the assessed times (*Figure 7—figure supplement 1*). Simultaneously, a separate group of flies was entrained for 3 days to the same 12:12 LD conditions and then placed in free-running conditions in constant darkness (DD) to assess the strength of the endogenous clock on these flies. While no significant effect was found in their periodicity, the rhythmicity levels of the flies overexpressing *gypsy* were affected compared to those of their age-matched controls. Also, 84–97% of the control flies were rhythmic at all the assessed ages, whereas only 73–77% of the flies with ectopic expression of the TE

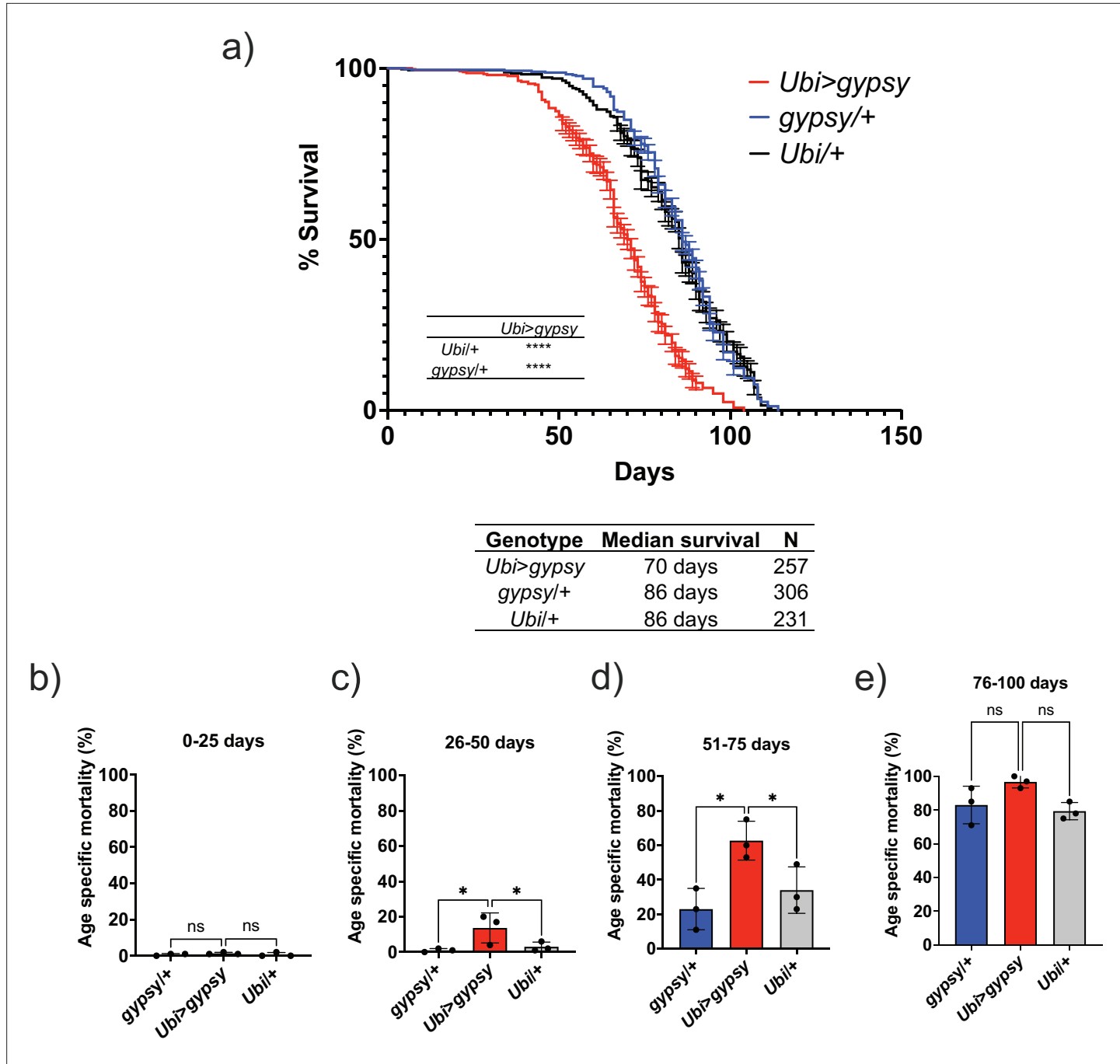

**Figure 4.** Ectopic *gypsy* expression decreases life span during old age. (**a**) Survival curves of male flies expressing *gypsy* under the control of *Ubiquitin* GAL4 (red) and the parental controls: *gypsy/+* (blue) and *Ubi/+* (black). Data represents three biological replicates (independent cohorts done at different times of year), error bars SE. p-value<0.0001, log-rank test. (**b–e**) Data are represented as means ± SD (individual measurements are shown as dots, age-specific mortality was calculated for each cohort independently). One-way ANOVA: adjusted p-value 26–50 days (*gypsy/+* 0.047, *Ubi/+* 0.047), adjusted p-value 51–75 days (*gypsy/+* 0.015, *Ubi/+* 0.030), ns, not significant.

The online version of this article includes the following figure supplement(s) for figure 4:

**Figure supplement 1.** Survival curves of female flies expressing *gypsy* under the control of *Ubiquitin* GAL4 (red) and the parental controls: *gypsy/+* (blue) and *Ubi/+* (black).

**Figure supplement 2.** Male vs. female parental UAS *gypsy* control.

**Figure supplement 3.** Head vs. body *gypsy* parental control.

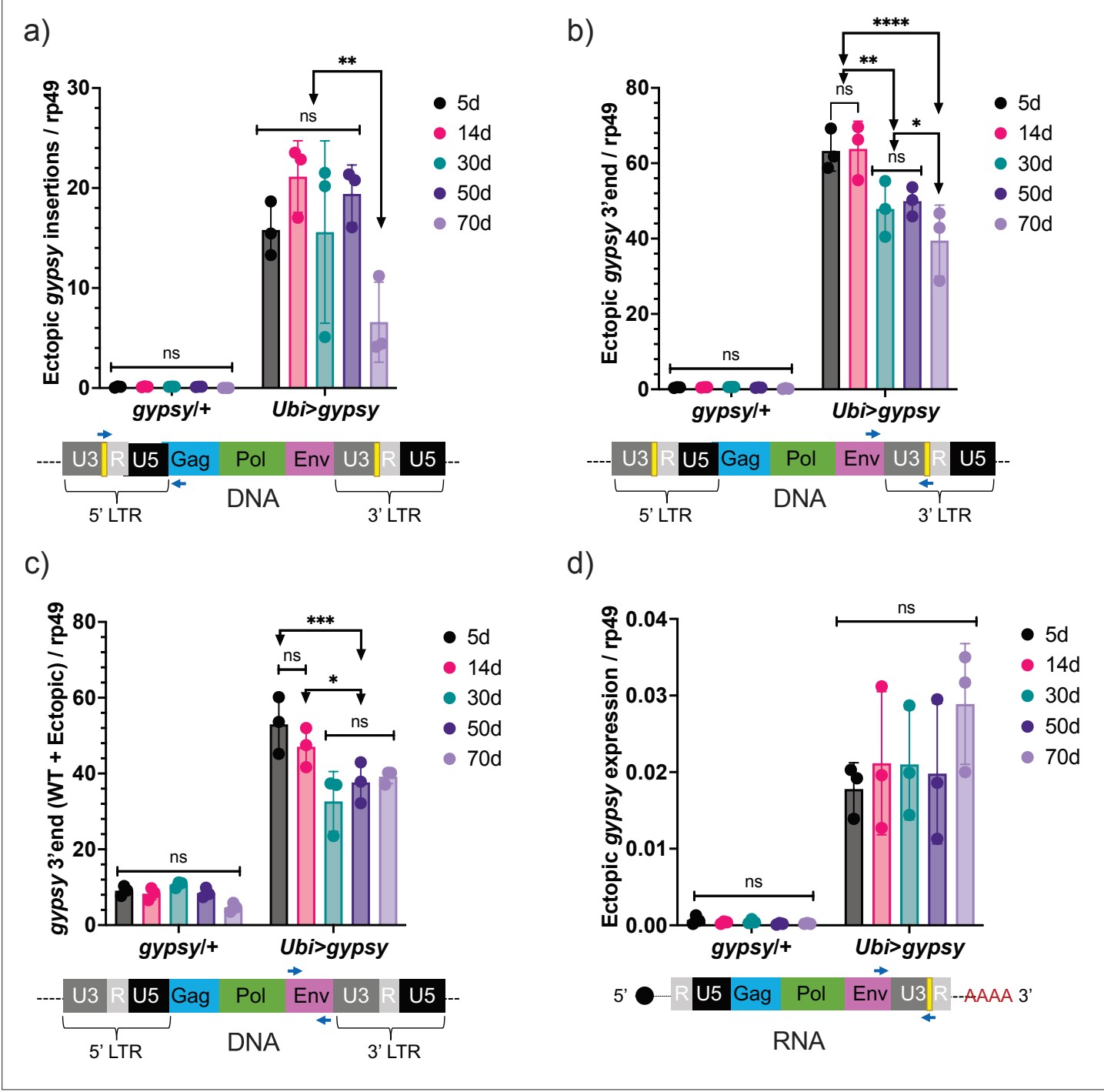

**Figure 5.** Ectopic *gypsy* DNA does not increase with age. (**a–c**) gDNA qPCR of male flies at different ages (5 days, 14 days, 30 days, 50 days, and 70 days). Data are represented as means ± SD (three biological replicates, each dot is a pool of 10 flies). Two-way ANOVA, ****adjusted p-value<0.0001, **adjusted p-value<0.01, *adjusted p-value<0.05. (**a**) Ectopic *gypsy* provirus junctions are detected. (**b**) 3' end of ectopic *gypsy* fragments are detected. (**c**) Wildtype (WT) and ectopic *gypsy env* fragments are detected. (**d**) RT-qPCR of male flies at different ages (5 days, 14 days, 30 days, 50 days, and 70 days). 3' end of ectopic *gypsy* transcript is detected. Data are represented as means ± SD (three biological replicates, each dot is a pool of five flies). Two-way ANOVA, not significant (ns).

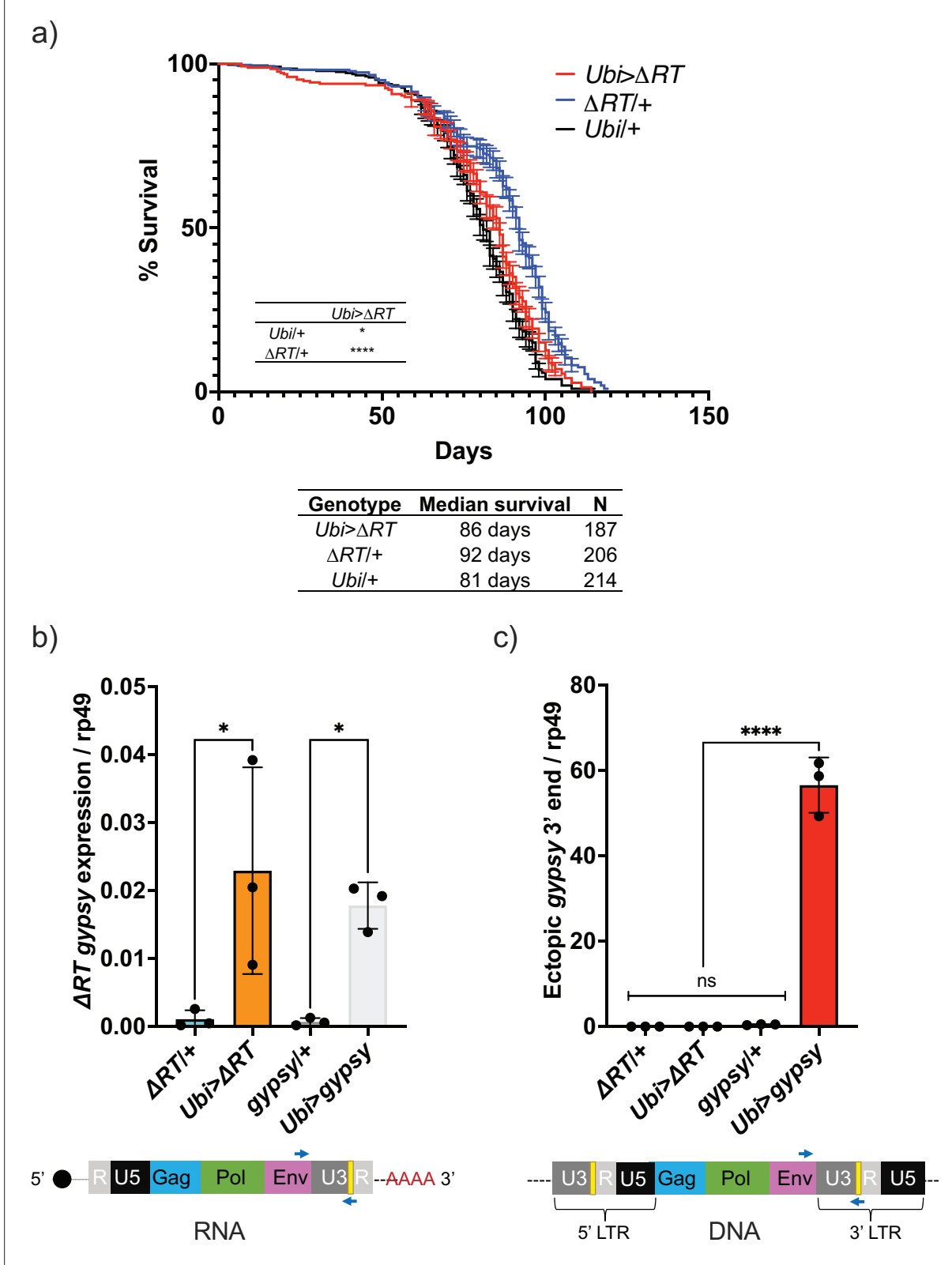

**Figure 6.** Decrease in life span requires reverse transcriptase activity. (**a**) Survival curves of male flies expressing *ΔRT gyps* under the control of *Ubiquitin* GAL4 (red) and the parental controls: *ΔRT gypsy/+* (blue) and *Ubi/+* (black). Data represents two biological replicates (staggered independent cohorts), error bars SE. p-Value<0.0001, *p-value=0.016, log-rank test. (**b, c**) Data are represented as means ± SD (three biological replicates, each dot is a pool of five flies). *gypsy/+* and *Ubi>gypsy* data are replotted from *Figure 2b and c*, respectively. (**b**) RT-qPCR of 5-day-old males. 3' end of ectopic *gypsy*

*Figure 6 continued*

transcript is detected. One-way ANOVA, *ΔRT/+* vs. *Ubi>ΔRT* \* adjusted p-value=0.012, *gypsy/+* vs. *Ubi>gypsy* \* adjusted p-value=0.029. (**c**) gDNA qPCR of 5-day-old males. 3' end of ectopic *gypsy* fragments is detected. Ordinary one-way ANOVA, \*\*\*\*adjusted p-value<0.0001.

remain rhythmic (*Figure 7b*). Indeed, these flies also display a significant decline in their rhythmicity index (RI) throughout the assessed ages that does not happen in the control flies (*Figure 7c*).

Finally, we also tested the ability of the flies with ectopic *gypsy* expression to move vertically when startled by performing a negative geotaxis assay. Briefly, we tapped the flies and recorded their location within a graduated cylinder for the following 15 s. We focused on the UAS parental control to select a height and time threshold optimal across different experimental ages (*Figure 7—figure supplement 2*). As a result and to capture possible subtle differences between the young flies and get a robust data capture even in older ages, we focused on the percentage of flies that climbed above the 7.5 cm threshold after 5 and 10 s. We examined the climbing ability of the active *gypsy* flies at four different ages (7 days, 14 days, 35 days, 56 days). As expected, we observed a decay of negative geotaxis with age. However, we did not detect any accelerated decay in the *gypsy* expressing flies when compared to parental controls (*Figure 7—figure supplement 3*).

Having created a system in which we can regulate the activity of a particular TE (*gypsy*) and monitor the effects of its accumulation upon life span and other aging hallmarks, we went back to our original question regarding the role of dFOXO against the negative effects of TE expression. The *Ubi*-GAL4 and UAS-*gypsy* lines were crossed to a UAS-*FOXO* line (*Slack et al., 2011*) to increase *dFOXO* expression in the animals. This leads to a 32% increase in the level of *dFOXO* mRNA in these animals (*Figure 8a*). We find that increased expression of *dFOXO* can rescue the effect on life span due to *gypsy* activity. The 19% decrease in life span reported in *Figure 4a* is reduced to 8% when compared to the UAS control and the life span matches the GAL4 control (*Figure 8b*). Importantly, the effect is not simply due to a decrease in ectopic *gypsy* mRNA expression as the transcript remains significantly induced in the *dFOXO*-overexpressing flies (compare *Figure 2b* with *Figure 8c*).

We next sought to determine whether dFOXO overexpression was affecting the aging phenotypes of the active *gypsy* flies. We tested the paraquat sensitivity of the 5-day-old active *gypsy* flies with dFOXO overexpression (*Figure 8d*). The survival curve indicates the dFOXO overexpression rescues the paraquat sensitivity previously seen in 5-day-old animals. We also measured the relative number of integrations of 5-day-old active *gypsy* flies with dFOXO overexpression (*Figure 8e*). We find that there is a significant decrease in gypsy insertions when dFOXO is overexpressed. Thus, dFOXO is able to rescue the effects of active *gypsy* expression despite having little effect on the steady-state expression level of the ectopic gypsy.

## Discussion

Aging can be described as a systemic breakdown due to the accumulation of different stress conditions (*López-Otín et al., 2013*). The stress response transcription factor FOXO can promote longevity by helping the cell respond to a myriad of conditions: oxidative stress, heat shock, virus infection, and defects in protein homeostasis to name a few (*Donovan and Marr, 2016*; *Martins et al., 2016*; *Spellberg and Marr, 2015*). Whether FOXO can protect from the detrimental effects of TE activity on life span is an open question.

To begin to investigate this question, we measured TE expression with age in dFOXO null and isogenic wt flies. Previous studies of wt animals report both increased and decreased TE mRNA levels with age (*Chen et al., 2016*; *LaRocca et al., 2020*). In one study of the female fat body (5 days vs. 50 days), researchers observed significant increases in 18 TE expression and decreases in 18 TEs out of the 111 detected (*Chen et al., 2016*). While the total number of TE with detectable mRNA changes is lower in our whole animal study, the fact that we detect both increases and decreases in the wild-type animals is consistent with the fat body study. In the genetic background used here, we find that 18 TEs have increased mRNA levels in dFOXO null flies, while only 2 TEs showed increased mRNA in wt. Surprisingly, no TE expression is significantly decreased with age in the dFOXO null flies. This indicates an increased TE load in aged dFOXO null animals. Furthermore, it suggests that the FOXO null animals are deficient in mounting a response to restrict TE expression.

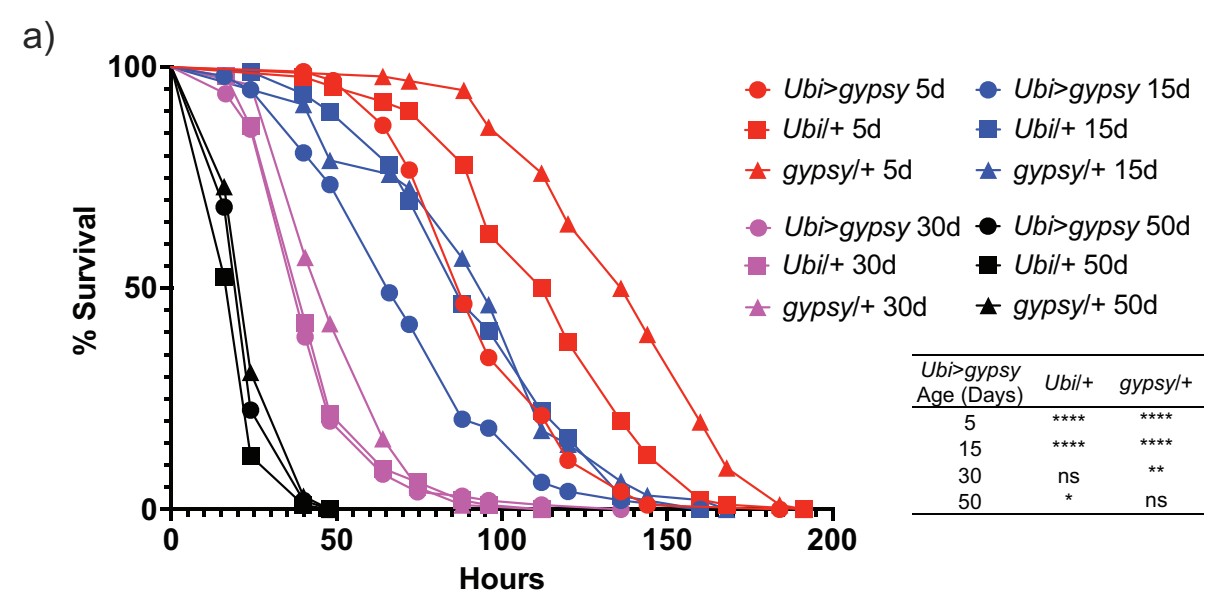

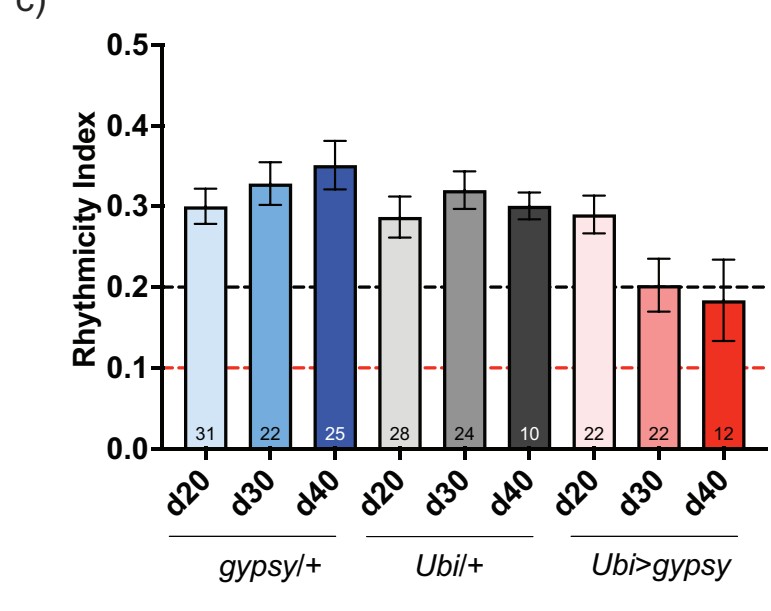

**Figure 7.** *gypsy* activity accelerates a subset of aging phenotypes. (**a**) Survival curves of male flies after exposure to 20 mM paraquat at different ages (red = 5 days, blue = 15 days, magenta = 30 days, and black = 50 days). Experimental flies expressing *gypsy* under the control of *Ubiquitin* GAL4 are represented as circles. The parental controls *Ubi/+* and *gypsy/+* are denoted squares and triangles, respectively. Statistical significance is indicated on graph. *p-value=0.013, **p-value=0.006, p-value<0.0001, log-rank test. (**b**) Summary table of the percentage of rhythmic flies (R%) and their period in

*Figure 7 continued on next page*

*Figure 7 continued*

DD. (**c**) Rhythmicity levels of each genotype across the assessed ages (± SEM). *gypsy*/+ control, *Ubi>gypsy* experimental, and *Ubi*/+ control flies are represented in blue, red, and gray, respectively. The black and red dotted lines mark the thresholds between what are considered highly (rhythmicity index [RI] > 0.2) and weakly rhythmic, and arrhythmic flies (RI < 0.1). The number at the bottom of each column indicates the n.

The online version of this article includes the following figure supplement(s) for figure 7:

**Figure supplement 1.** Total activity counts of each genotype *per* day under 12:12 light–dark (LD) conditions.

**Figure supplement 2.** Negative geotaxis threshold optimization.

**Figure supplement 3.** Negative geotaxis.

Each fly strain has a unique set of TEs that are capable of being transcribed, which is likely due to differences in the TE landscape such as the number and location of individual TE copies in the genome (*Rahman et al., 2015*). We observed this difference in expressed TEs even when comparing expression in young flies (*Figure 1e*). This agrees with previous work showing supposedly isogenic stocks of *D. melanogaster* can have very different TE landscapes (*Rahman et al., 2015*). This difference in TE content and expression between our wt and dFOXO null strain makes it difficult to determine the effect of FOXO on individual endogenous TEs.

We created the UAS-*gypsy* system to circumvent the difference in TE landscapes and simultaneously perform a direct assay to determine whether TE activity in somatic tissue can be a causative agent of mortality and aging-associated phenotypes. We chose the *gypsy* TE as our model for several reasons. First, because previous work has shown that *gypsy* insertions increase during aging, so this retrotransposon is relevant in the natural condition (*Chang et al., 2019a*; *Li et al., 2013*). Second, a full-length clone of *gypsy* is available and it has been shown to be active and capable of transposition (*Bayev et al., 1984*). Lastly, the presence of only one full-length copy of *gypsy* in the *D. melanogaster* reference genome (*Kaminker et al., 2002*) suggests a low copy number and mitigates possible unintended trans effects between the endogenous and ectopic TE. To further separate our ectopic TE, a unique 3′ sequence tag was inserted. This allows the detection and differentiation of *gypsy* mRNA and DNA content derived from our ectopic *gypsy* element.

The presence of the sequence tag in the newly formed 5′ LTR of new ectopic *gypsy* insertions allowed us to use our targeted sequencing approach to map a large number of individual new *gypsy* insertions derived from the UAS-*gypsy* synthetic TE. The target site duplication matched the site of endogenous *gypsy* elements (*Dej et al., 1998*), indicating that the UAS-*gypsy* element likely goes through a replication cycle identical to the natural *gypsy* element. The insertions seem to be evenly distributed and map largely to intronic sequences. This may reflect the fact that we can only recover insertions that do not have a dramatic effect on cell growth. Genomes that suffer insertions disrupting genes that are required for cell viability will be lost from the population and will not be recovered. This approach shows that the UAS-*gypsy* element can make an active transposon that can insert at sites across the genome with little chromosomal bias.

The induction of TE activity in somatic tissue resulted in a reduction in *D. melanogaster* life span when compared to parental controls and a significant increase in mortality in middle-aged animals. There was a 19% decrease in the life span of male animals. Interestingly, the mortality effects of the active TE are only evident in relatively aged animals despite the fact that the retrotransposon is active during early life. This suggests that the young animals can tolerate expression and insertion of the TE. It is only when the animals begin to age that TE expression becomes a burden and takes a toll. Perhaps it is the combination of the other metabolic and physiological effects of aging with the TE activity that is detrimental. Coincidentally, this is also the time frame that endogenous TE become expressed during normal aging (*Li et al., 2013*; *Yang et al., 2022*). However, as the ubi-GAL4 driver is active throughout the life cycle of the fly, we cannot currently define the exact time window when TE activity affects life span and aging phenotypes.

The detectable ectopic *gypsy* insertions were quantified to determine whether an increase in detected insertions in the active *gypsy* flies would correlate with the decrease in life span. Unexpectedly, detectable insertions do not seem to increase with age, despite constant (although much more variable) expression of the transgene as the animals age. In fact, the oldest animals have the lowest detectable insertions. This finding was measured through three different approaches (the 5′ new insertion junction, the 3′ fragment of ectopic *gypsy*, and the wildtype *gypsy* env gene). All three

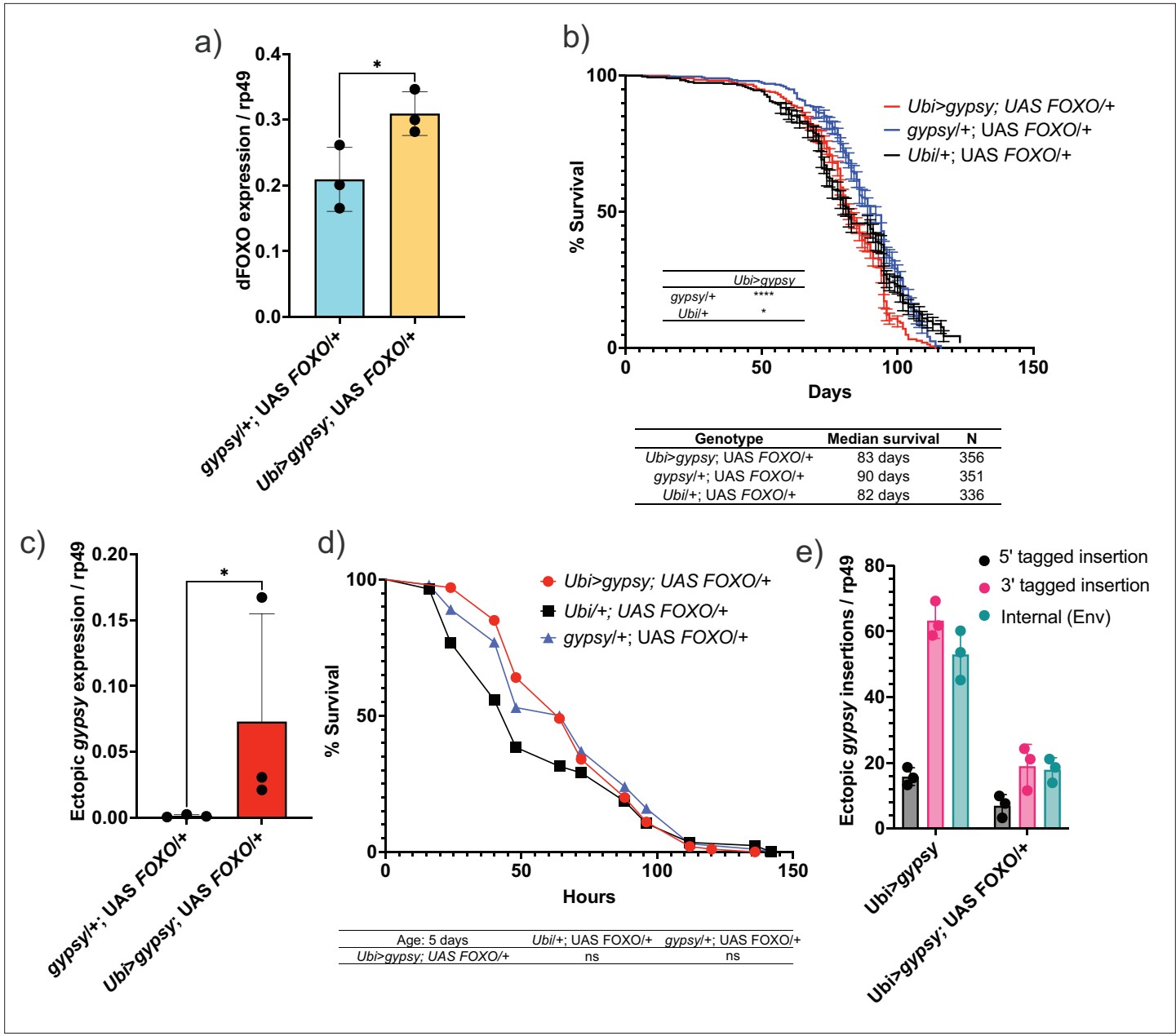

**Figure 8.** Increasing Forkhead-box O family (FOXO) activity can rescue longevity effect. (**a**) *dFOXO* exon 8 is detected. RT-qPCR of 5-day-old males. Data are represented as means ± SD (three biological replicates, each dot is a pool of five flies). One-tailed *t*-test, *p-value=0.021. (**b**) Survival curves of male flies expressing *gypsy* and *dFOXO* under the control of *Ubiquitin* GAL4 (red line) and the parental controls: *Ubi/+*; UAS-*FOXO/+* (black) and *gypsy/+*; UAS-*FOXO/+* (blue). Data represents two biological replicates, error bars SE. *p-value=0.018, p-value<0.0001, log-rank test. (**c**) 3' end of ectopic *gypsy* transcript is detected. RT-qPCR of 5-day-old males. Data are represented as means ± SD (three biological replicates, each dot is a pool of five flies) One-tailed Mann–Whitney test, *p-value=0.05. (**d**) Survival curves of male flies exposed to 20 mM paraquat expressing *gypsy* and *dFOXO* under the control of *Ubiquitin* GAL4 (red line) and the parental controls: *Ubi/+*; UAS-*FOXO/+* (black) and *gypsy/+*; UAS-*FOXO/+* (blue). Log-rank test result for curve comparisons is shown below the curve; *Ubi>gypsy*; UAS *FOXO/+n* = 100, *Ubi/+*; UAS *FOXO/+n* = 86, *gypsy/+*; UAS *FOXO/+n* = 100. (**e**) gDNA qPCR of male flies at 14 days. Data for 14-day *Ubi>gypsy* from ***Figure 5*** are replotted for comparison. Data are represented as means ± SD (three biological replicates, each dot is a pool of 10 flies).

approaches agree and show a consistent decrease with age of ectopic *gypsy* DNA content. This suggests that unknown mechanisms are acting to clear or at least prevent an increase in TE insertion load. Whether it is at the level of cellular loss or DNA repair remains to be determined.

By using a UAS-*gypsy* strain with a defective RT, we were able to determine that a functional RT is needed for the TE effect on life span. Previous studies have suggested the need for RT to see

detrimental TE effects (*Gorbunova et al., 2021*). The reverse transcriptase inhibitor 3TC extends the life span of a *Dcr-2* null fly strain, which has an increase in TE expression (*Wood et al., 2016*). The need for a functional RT to decrease life span implicates the DNA synthesis step as being detrimental. Because this also prevents downstream steps such as gene disruption through integration or DNA damage from incomplete integration, it is not clear what step beyond DNA synthesis is most detrimental. Future experiments using this approach with integrase mutants of *gypsy* may help to answer this question.

The decrease in life span also opened the question of whether the active *gypsy* flies were aging more rapidly and whether accelerated aging phenotypes could be detected. Criteria to determine premature aging and distinguish it from other causes can be summarized as follows *Salk, 2013*. It must first be determined that the increase in mortality at younger ages does not alter the shape of the survival curve. An altered shape of the survival curve indicates that the health of the subjects was compromised and the increase in mortality could be due to unforeseen disease or other factors apart from natural aging processes (*Piper and Partridge, 2016*). Secondly, a proportional progression of all aging phenotypes without induction of disease must also be observed. The shape of the survival curve for the active *gypsy* flies indicated that the flies were healthy and thus the increase in mortality was not due to disease or external factors. The health of the active *gypsy* flies in the face of their increased mortality led us to test whether aging-associated phenotypes might be proportionally progressing in them. To determine whether this was the case, four aging phenotypes were measured to try to detect whether an acceleration in phenotype development with respect to the parental controls was occurring in the active *gypsy* flies.

Not all phenotypes responded the same. Only two out of the four phenotypes assayed show an accelerated decay. An active *gypsy* accelerates the decrease in resistance to paraquat of aging flies. Interestingly, this finding parallels the hypersensitivity to oxidative stress observed when *Dcr-2* is mutated (*Lim et al., 2011*). Perhaps most dramatically, 5-day-old animals with an active *gypsy* have the oxidative stress resistance of a 15-day-old animal. The animals' rhythmicity is also impacted. In active *gypsy* flies, rhythmicity decays at a slightly faster rate with most animals becoming only weakly rhythmic by day 30. Other measures of circadian behavior such as total activity and period did not change with an active TE. The decay of locomotor activity also occurred at a rate similar to controls, indicating that not all aging phenotypes show an accelerated decay in response to an active TE. The absence of a locomotor defect also parallels what was observed in the *Dcr-2* null fly strain (*Lim et al., 2011*). The distinct effects an active TE can generate on the aging phenotypes examined imply that the hallmarks of aging are not uniformly affected, and different aging processes might originate or impact a unique subset of hallmarks. We find that an active TE can accelerate a subset of aging phenotypes and provide evidence that TEs are not merely bystanders in the aging process and can behave as causative agents once they are active.

The development and use of a controllable TE expression system with a direct detrimental effect on longevity allowed us to assay whether increasing the activity of the transcription factor dFOXO played a role in promoting longevity in the face of an active TE. We find that mild overexpression of *dFOXO* can rescue the life span defect in the active *gypsy* flies. Though the UAS-*gypsy* was active, the decrease in life span was almost completely rescued, highlighting dFOXO's ability to prolong life span in the face of TE activity.

In an attempt to determine how dFOXO expression was protective against the active TE, we assayed paraquat resistance and the relative ectopic *gypsy* integration in dFOXO-overexpressing flies. We find that dFOXO overexpression rescues the paraquat sensitivity in the animals with active *gypsy*. This is perhaps not surprising because we and others have previously shown that FOXO responds to paraquat-induced oxidative stress (*Chang et al., 2019b*; *Donovan and Marr, 2016*; *Wang et al., 2005*). We also measured the relative integration number of 5-day-old animals expressing UAS-*gypsy* and overexpressing dFOXO. Surprisingly, despite the fact that the UAS-*gypsy* mRNA is expressed at comparable levels (compare *Figure 8c* and *Figure 2b*), animals overexpressing dFOXO have a lower number of insertions. Overexpression of dFOXO is both protecting genome integrity and fending of the oxidative stress caused by the active *gypsy* TE.

The FOXO family of transcription factors regulate many pathways that could be reducing the detrimental effects of increased TE expression such as the DNA damage response pathway, antioxidant pathways, and proteostasis pathways (*Gui and Burgering, 2021*; *Webb et al., 2016*). In addition,

dFOXO activates pathways that are established antagonists to the detrimental effects of TEs such as the RNAi pathway (*Spellberg and Marr, 2015*). The sum of these responses would enhance the ability of dFOXO to combat the detrimental effect on life span caused by an active TE and suggests a potential new role for dFOXO in its vast repertoire to promote longevity (*Martins et al., 2016*).

# Materials and methods

## Key resources table

| Reagent type (species) or resource | Designation | Source or reference | Identifiers | Additional information |
|---|---|---|---|---|
| Strain, strain background (*Drosophila melanogaster*) | Wildtype control; wt; wDAH | Doi: 10.1111/j.1474-9726.2011.00707.x; *Slack et al., 2011* | | |
| Strain, strain background (*D. melanogaster*) | w1118 | Kadener Lab, Brandeis University | | |
| Genetic reagent (*D. melanogaster*) | dFOXO null; wDAH Δ94; dFOXO deletion animals | Bloomington Drosophila Stock Center | BDSC: 42220 | |
| Genetic reagent (*D. melanogaster*) | UAS-*gypsy* TAG; UAS-*gypsy* | This paper | | See 'Fly stocks, *D. melanogaster* husbandry, and constructs' |
| Genetic reagent (*D. melanogaster*) | UAS-ΔRT; RT deletion | This paper | | See 'Fly stocks, *D. melanogaster* husbandry, and constructs' |
| Genetic reagent (*D. melanogaster*) | *Ubi*-Gal4 | Bloomington *Drosophila* Stock Center | BDSC: 32551 | |
| Genetic reagent (*D. melanogaster*) | UAS-FOXO | Bloomington *Drosophila* Stock Center | BDSC: 42221 | |
| Sequence-based reagent | Tag-Provirus junction_F | This paper | PCR primers | GCCAAGCTCAGAATTAACCC |
| Sequence-based reagent | Tag-Provirus junction_R | This paper | PCR primers | TGGTGGGTTCAGATTGTTGG |
| Sequence-based reagent | *gypsy* env – Tag_F | This paper | PCR primers | TACAGCGCACCATCGATACT |
| Sequence-based reagent | *gypsy* env – Tag_R | This paper | PCR primers | GTGAGGGTTAATTCTGAGCTTG |
| Sequence-based reagent | *gypsy* env_F | This paper | PCR primers | CTCTGCTACACCGGATGAGT |
| Sequence-based reagent | *gypsy* env_R | This paper | PCR primers | AGTATCGATGGTGCGCTGTA |
| Sequence-based reagent | Rp49_F | This paper | PCR primers | CCACCAGTCGGATCGATATGC |
| Sequence-based reagent | Rp49_R | This paper | PCR primers | CTCTTGAGAACGCAGGCGACC |
| Sequence-based reagent | FOXO_F | This paper | PCR primers | CACGGTCAACACGAACCTGG |
| Sequence-based reagent | FOXO_R | This paper | PCR primers | GGTAGCCGTTTGTGTTGCCA |
| Sequence-based reagent | Primers for NGS | This paper | | See *Table 2* |
| Commercial assay or kit | TruSeq RNA Library Prep Kit v2 | Illumina | RS-122-2001 | |

*Continued on next page*

*Continued*

| Reagent type (species) or resource | Designation | Source or reference | Identifiers | Additional information |
|---|---|---|---|---|
| Chemical compound, drug | Tri Reagent | Molecular Research Center | TR 118 | |
| Chemical compound, drug | Paraquat | Sigma | 856177-1G | |
| Software, algorithm | FASTQ Groomer | Galaxy web platform (usegalaxy.org) | | See 'Bioinformatics' |
| Software, algorithm | FastQC | Galaxy web platform (usegalaxy.org) | | See 'Bioinformatics' |
| Software, algorithm | RNA STAR aligner | Galaxy web platform (usegalaxy.org) | | See 'Bioinformatics' |
| Software, algorithm | Trimmomatic | Galaxy web platform (usegalaxy.org) | | See 'Insertion mapping' |
| Software, algorithm | Barcode Splitter | Galaxy web platform (usegalaxy.org) | | See 'Insertion mapping' |
| Software, algorithm | cutadapt | Galaxy web platform (usegalaxy.org) | | See 'Insertion mapping' |
| Software, algorithm | Bowtie2 | Galaxy web platform (usegalaxy.org) | | See 'Insertion mapping' |
| Software, algorithm | DEBrowser | *Kucukural et al., 2019* | | See 'Bioinformatics' |
| Software, algorithm | weblogo3 | *Crooks et al., 2004* | | See 'Insertion mapping' |
| Software, algorithm | GraphPad Prism | GraphPad Software | Version 9.4.0 | |
| Software, algorithm | MATLAB: Vecsey's SCAMP | MathWorks, *Donelson et al., 2012* | | |
| Other | M-270 Dynabeads | Thermo Fisher | Cat# 65305 | Magnetic resin |
| Other | NextSeq 500 | Illumina | Cat# SY-415-1001 | Illumina sequencing platform reagents |
| Other | *D. melanogaster* genome (dm6) | Galaxy web platform (usegalaxy.org) | | See 'Bioinformatics' |
| Other | TE consensus FASTA | TIDAL Database, *Rahman et al., 2015* | | See 'Bioinformatics' |
| Other | Flybase genespan | Flybase.org Genome Browser | | See 'Insertion mapping' |
| Other | Intron annotation for Ensembl | Ensembl | intron_annotation_*Drosophila*_ensemblv84.bed | See 'Insertion mapping' |

## Fly stocks, *D. melanogaster* husbandry, and constructs

*D. melanogaster* stocks and experimental flies were maintained at 25°C with a 12 hr light/dark cycle at controlled relative humidity. Male flies were used for this study. The fly strains used for RNA-seq were dFOXO null ($w^{DAH\ \Delta 94}$) and its isogenic wildtype control $w^{DAH}$, both have been previously described (*Spellberg and Marr, 2015*). The UAS-*gypsy* TAG fly strain was created by modifying the plasmid

pDM111, which contains an active copy of *gypsy* (*Bayev et al., 1984*; a generous gift from the Corces lab). The white marker from pTARG (*Egli et al., 2006*) and a phiC31 attB site were added to the plasmid and a unique sequence was inserted in the *gypsy* 3' LTR. The 5' LTR of *gypsy* was precisely replaced with the UAS promoter from pUAST (*Brand and Perrimon, 1993*) such that the start of transcription matched the start for the *gypsy* LTR. For the RT deletion, the UAS-*gypsy* parent plasmid was cut with AflII to create an in-frame deletion of most of the RT from the *gypsy* polyprotein (*Marlor et al., 1986*). The constructs were sent to BestGene Inc (Chino Hills, CA) for injection into *D. melanogaster*. Transgenes were integrated into the VK37 (*Venken et al., 2006*) attP site (BDSC 9752) using PhiC31 integrase and balanced. The line was then extensively backcrossed into the $w^{1118}$ background. The *Ubi*-GAL4 and UAS-FOXO fly strains were obtained from Bloomington (32551 and 42221, respectively) and crossed for at least five generations into the $w^{1118}$ lab stock. The following strains were generated by crosses: $w^{1118}$; *Ubi >gypsy* ($w^{1118}$; UAS-*gypsy* mated to $w^{1118}$; *Ubi*-GAL4), $w^{1118}$; *Ubi>ΔRT* ($w^{1118}$; *ΔRT* mated to $w^{1118}$; *Ubi*-GAL4), $w^{1118}$; *Ubi/+*; UAS-FOXO/+ ($w^{1118}$; *Ubi*-GAL4;+/TM3 mated to $w^{1118}$; UAS-FOXO), $w^{1118}$; *Ubi/UAS-gypsy*; UAS-FOXO/+ and $w^{1118}$; UAS-*gypsy/+*; UAS-FOXO/+ ($w^{1118}$; *Ubi/+*; UAS-FOXO/+ mated to $w^{1118}$; UAS-*gypsy*). All flies used throughout our experimental procedures were placed in the *w1118* genetic background.

## RNA-seq

Total RNA from the whole body of 10 male $w^{DAH}$ and dFOXO null ($w^{DAH\,Δ94}$) flies was extracted with TRI Reagent at 5–6 days and 30–31 days old according to the manufacturer's protocol (Molecular Research Center, Inc, Cincinnati, OH). To generate RNA-seq libraries, 1 μg of total RNA was used as input for the TruSeq RNA Library Prep Kit v2 (Illumina, Inc, San Diego, CA) and the manufacturer's protocol was followed. Libraries were sequenced on an Illumina NextSeq 500 in 1 × 75 bp mode. Three individually isolated biological replicates were sequenced for each condition.

## Bioinformatics

RNA-seq fastQ files were uploaded to the public server usegalaxy.org and processed at the Galaxy web platform (*Afgan et al., 2018*). The tools FASTQ Groomer (*Blankenberg et al., 2010*) and FastQC (*Andrews, 2010*) were used for library quality control. To obtain gene counts, the RNA STAR aligner (*Dobin et al., 2012*) was used to map the sequencing data to both the *D. melanogaster* genome (dm6) and to a TE consensus FASTA file with 176 *Drosophila* TE. The R package DEBrowser (*Kucukural et al., 2019*) was used for the following procedures: to filter the counts (1 count per million [CPM] in at least 11/12 libraries), 105 TEs passed filtering, calculate differential expression and statistical significance with DESeq2 (parameters: 5% false discovery, 1.5-fold, local, no beta prior, LRT), and generate volcano and scatter plots.

## Life span assays

Life span assays were performed as previously described (*Linford et al., 2013*). Briefly, newly eclosed flies were mated for 48 hr, sorted by gender, and kept at a standard density of 15 flies per vial. Flies were transferred to fresh food every 2–3 days and mortality was recorded. Three independent biological replicates of at least 100 flies were performed at different times for *Ubi>gypsy*. The age-specific mortality rate was calculated by dividing the deaths occurred in a given age group by the size of the population in which the deaths occurred (*Principles of Epidemiology, 2021*). Two independent biological replicates were performed at different times for *Ubi>gypsy*; UAS-FOXO/+ and *Ubi>ΔRT*.

**Table 2.** Primers.

| Target | Forward | Reverse |
|---|---|---|
| Tag-Provirus junction | GCCAAGCTCAGAATTAACCC | TGGTGGGTTCAGATTGTTGG |
| *gypsy* env – Tag | TACAGCGCACCATCGATACT | GTGAGGGTTAATTCTGAGCTTG |
| *gypsy* env | CTCTGCTACACCGGATGAGT | AGTATCGATGGTGCGCTGTA |
| Rp49 | CCACCAGTCGGATCGATATGC | CTCTTGAGAACGCAGGCGACC |
| FOXO | CACGGTCAACACGAACCTGG | GGTAGCCGTTTGTGTTGCCA |

Kaplan–Meier survival curves were generated with GraphPad Prism version 9 (GraphPad Software, San Diego, CA, https://www.graphpad.com) and analyzed with the log-rank test.

## RT-qPCR and genomic DNA qPCR

Total RNA or DNA was extracted from the whole body of 5–10 male flies. RNA was extracted with the TRI Reagent according to the manufacturer's protocol (Molecular Research Center, Inc) and genomic DNA as described previously (*Aljanabi and Martinez, 1997*). cDNA for RT-qPCR was synthesized as previously described (*Olson et al., 2013*). RT-qPCR and gDNA qPCR were performed as follows: for a 10 µl reaction 2 µl of cDNA or 50 ng of DNA were used as template and assayed with SYBR green with the primers in *Table 2*. For all experiments, three biological replicates were assayed for each condition and the relative expression was calculated as a fraction of the housekeeping gene Rp49.

## Oligo adenylation

MB1192 was adenylated using recombinant MTH ligase (*Zhelkovsky and McReynolds, 2011*). Then, 200 pmol of oligo was adenylated in a 200 µl reaction (50 mM Tris 7.5, 10 mM $MgCl_2$, 0.1 mM EDTA, 0.1 mM ATP, 5 mM DTT) with 10 µl recombinant MTH ligase. The reaction was incubated at 65°C for 1 hr and terminated by incubation at 85°C for 15 min. Then, 20 µg of glycogen was added to the reaction followed by 500 µl ethanol. The oligo was placed at –20°C for 30 min and collected by centrifugation. The pellet was resuspended in 100 µl 6 M GuHCl, 50 mM Tris 6.8. Also, 400 µl of ethanol was added and the oligo was loaded onto a silica spin column (BioBasic PCR cleanup). The column was washed once with 80% ethanol and then centrifuged until dry. The adenylated oligo was eluted in 40 µl TE and quantitated by UV absorbance.

## NGS library preparation

To prepare the insertion libraries, 250 ng of genomic DNA was digested with the 4-base cutter MnlI (NEB, Ipswich, MA), overnight at 37°C according to the manufacturer's recommendations, to fragment the genome. Additionally, MnlI cuts the UAS-*gypsy* element 61 times including 17 bp from the 5′ end of the 3′ LTR to limit the recovery of the parental UAS-*gypsy* sequences. The fragmented DNA was purified using a silica-based PCR cleanup kit (Biobasic, Markham, Canada). A biotinylated primer (MB2640) annealing to the TAG sequence was annealed and extended with 20 cycles of linear amplification with pfuX7 (*Nørholm, 2010*) in a 100 µl reaction (20 mM Tris–HCl pH 8.8 at 25°C, 10 mM $(NH_4)2SO_4$, 10 mM KCl, 2 mM $MgSO_4$, 0.1% Triton X-100, 200 µM dNTPs, 0.5 µM primer, 5u pfuX7) The reaction was quenched by adding 400 µl TENI (10 mM Tris 8.0, 1 mM EDTA, 25 mM NaCl, 0.01% Igepal 630). Biotinylated products were purified using M270 Dynabeads (Thermo Fisher, Waltham, MA) by incubating the reaction for 30 min at room temperature followed by magnetic separation of bound DNA. The beads were washed three times with TENI. Single-stranded biotinylated DNA was purified by incubating the beads with 0.15 N NaOH for 15 min at room temperature. Beads were washed once more with 0.15 N NaOH to remove the non-biotinylated DNA strand. Beads were neutralized by washing two times in TENI and transferred to a new tube. The adenylated MB1192 oligo was ligated to the biotinylated ssDNA on the dynabeads using MTH ligase (*Torchia et al., 2008*). Beads were resuspended in a 60 µl reaction containing 10 mM HEPES pH 7.4, 5 mM $MnCl_2$ and 60 pmol adenylated MB1192 oligo. Then, 5 µl recombinant MTH RNA ligase was added and the reaction was incubated at 65°C for 1 hr and terminated by incubation at 85°C for 15 min. The library was amplified by PCR using a primer to the ligated product (MB1019) and a nested primer containing a 5′ tail with Illumina sequences (MB2669). The individual libraries were amplified by PCR using primers containing attachment sequences (MB583) and barcodes (MB26673 or MB2674 or MB2675). Libraries were sequenced on an Illumina MiSeq using a PE 150 kit.

## Insertion mapping

To process the reads, the fastQ file was first separated by Illumina barcode. The individual libraries were processed using Galaxy (*Afgan et al., 2018*). First MB2667 primer sequences were removed from read 1 using Trimmomatic (*Bolger et al., 2014*). Reads containing the *gypsy* LTR were separated using Barcode Splitter. The remaining LTR sequences were removed using cutadapt (*Martin, 2011*). The reads were filtered for size and then aligned to the UAS-*gypsy* construct. Then, 20–35% of the reads mapped to the original parental UAS-*gypsy*. Unaligned reads were then aligned to the

*Drosophila* genome (dm6) using Bowtie2 (*Langmead and Salzberg, 2012*). PCR duplicates were removed, and the insertion sites were compared to the flybase genespan annotation to identify all insertions in transcribed regions. Insertion sites were compared to the intron annotation for Ensembl for identifying insertions in introns. The reverse complement of the first 22 nucleotides of the deduplicated insertion sites were used to determine the probability of finding each nucleotide at each position using weblogo3 (*Crooks et al., 2004*).

## Paraquat stress assay

Male flies generated and reared in the same conditions as life span assay flies were fed 20 mM paraquat at 5, 15, 30, or 50 days. Briefly, at the specified time point flies were starved for 3–4 hr prior to transfer into a minimal food medium (5% sucrose, 2% agar, water) containing 20 mM paraquat. Mortality was recorded in 8 and 16 hr intervals. Approximately 100 flies were used per genotype/ time point. Kaplan–Meier survival curves were generated with GraphPad Prism version 9 (GraphPad Software) and analyzed with the log-rank test.

## Negative geotaxis assay

A negative geotaxis assay based on previously established protocols (*Gargano et al., 2005*; *Madabattula et al., 2015*; *Tuxworth et al., 2019*) was set up to assay experimental and control flies at 7, 14, 35, and 56 days. Briefly, male flies generated and reared in life span assay conditions were used. The same cohort was assayed for all the different time points. The day before the experiment, flies were transferred under light $CO_2$ to fresh food vials (10 flies per vial) and allowed to recover for at least 18 hr. Flies were then taken out of the incubator and allowed to acclimate to room temperature for 1 hr. They were then transferred to a 50 ml graduated glass cylinder (VWR, Radnor, PA) sealed with a cotton plug and allowed to acclimate for 1 min before starting trials. Trials were recorded on an iPhone X camera (Apple Inc, Cupertino, CA) placed 30 cm away from the recording spot. A trial consisted in tapping flies to the base of the cylinder and allowing them to climb for 20 s. They were then given 60 s to recover before starting the next trial. A total of five trials was performed for each vial. Ten vials were assayed per genotype (N = 100). Flies were then transferred to fresh food vials and returned to the incubator. Linear regression was performed in Prism (GraphPad Software, https:// www.graphpad.com) to determine the slope of the curves and any possible significant differences.

## Locomotor activity analysis

Flies were entrained in 12:12 LD conditions at 25°C. The locomotor activity of 20-, 30-, and 40-day-old male flies was recorded using the Trikinetics locomotor activity monitor (Waltham, MA). Two sets of experiments were conducted. On one set, flies were maintained in 12:12 LD throughout the whole length of the experiment (10–12 days). The other set was entrained in 12:12 for 3 days, followed by at least 5 days in constant darkness (DD). Quantification of total activity and the analysis of circadian rhythmicity strength and period were conducted in MATLAB using Vecsey's SCAMP (*Donelson et al., 2012*). Flies with a RI < 0.1 were classified as arrhythmic; ones with RI 0.1–0.2 as weakly rhythmic, while flies with RI > 0.2 were considered strongly rhythmic. Only the period of weakly or strongly rhythmic flies was included in the calculation of the free-running period for each genotype.

## Statistics

All statistical analyses, except RNA-seq, were conducted using GraphPad Prism version 9 for Mac (GraphPad Software, https://www.graphpad.com). Multiple comparisons in the ordinary one-way and two-way ANOVA were corrected by the two-stage linear step-up procedure of Benjamini, Krieger, and Yekutiel.

---

# Additional information

### Funding

| Funder | Grant reference number | Author |
| --- | --- | --- |
| National Institute on Aging | R21AG054724 | Joyce Rigal |

| Funder | Grant reference number | Author |
|---|---|---|
| National Institute on Aging | R01AG057700 | Sebastian Kadener |

The funders had no role in study design, data collection and interpretation, or the decision to submit the work for publication.

## Author contributions

Joyce Rigal, Conceptualization, Data curation, Formal analysis, Supervision, Validation, Investigation, Methodology, Writing - original draft, Project administration, Writing – review and editing; Ane Martin Anduaga, Conceptualization, Formal analysis, Investigation, Methodology, Writing – review and editing; Elena Bitman, Emma Rivellese, Investigation, Methodology, Writing – review and editing; Sebastian Kadener, Resources, Writing – review and editing; Michael T Marr, Conceptualization, Resources, Data curation, Formal analysis, Supervision, Funding acquisition, Investigation, Methodology, Project administration, Writing – review and editing

## Author ORCIDs

Ane Martin Anduaga http://orcid.org/0000-0003-2447-2195
Sebastian Kadener http://orcid.org/0000-0003-0080-5987
Michael T Marr http://orcid.org/0000-0002-7366-7987

## Decision letter and Author response

Decision letter https://doi.org/10.7554/eLife.80169.sa1
Author response https://doi.org/10.7554/eLife.80169.sa2

## Additional files

### Supplementary files

- Supplementary file 1. Total transposable element (TE) expression in wildtype (wDAH) flies.
- Supplementary file 2. Total transposable element (TE) expression in dFOXO null (Δ94) flies.
- MDAR checklist

### Data availability

Sequencing data have been deposited in GEO under accession code GSE205416.

The following dataset was generated:

| Author(s) | Year | Dataset title | Dataset URL | Database and Identifier |
|---|---|---|---|---|
| Rigal J, Marr MT | 2022 | Age dependent gene and transposon expression in male wildtype (wDAH) and FOXO deletion (wDAH Δ94) fly strains | https://www.ncbi.nlm.nih.gov/geo/query/acc.cgi?acc=GSE205416 | NCBI Gene Expression Omnibus, GSE205416 |

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
