## [Editor Report]

The work presented in this article clearly shows that the *Drosophila* FOXO gene when expressed can mitigate the effects of enhanced transposon activation on aging. Using the transposon gypsy under the expression of the UAS Gal4 the work shows convincingly that such expression decreases life span. The effects of such transposition are dependent upon reverse transcription and by an unknown pathway the FOXO can mediate the aging pathway(s) as the later protein seems not to be controlling transposon expression.

---

## [Decision Letter]

**Decision letter after peer review:**

Thank you for submitting your article "Artificially stimulating retrotransposon activity increases mortality and accelerates a subset of aging phenotypes in *Drosophila*" for consideration by *eLife*. Your article has been reviewed by 2 peer reviewers, and the evaluation has been overseen by a Reviewing Editor, Michael Botchan and Carlos Isales as the Senior Editor. The reviewers have opted to remain anonymous. The decision is to ask you and your colleagues to make revisions as suggested below.

Your revisions must address the major concerns of reviewer #2.

1) You need to better explain your proposal as to how dFOXO actually works in the aging system you use. Is it a direct effect upon transposition? A discussion or clarification of this would be important. Reviewer #1 also asked if there might be cis-acting binding sites for FOXO and this may be relevant.

2) You need to determine what the actual execution point during the life of the fly is when the effects of aging are manifest. Reviewer #2 points out that the experiments seem to not allow for the strong conclusions you have. Either modify with a clarification or address the point with a proviso.

*Reviewer #1 (Recommendations for the authors):*

This paper is interesting, well done and just the type of paper *eLife* should publish.

*Reviewer #2 (Recommendations for the authors):*

Since FOXO overexpression does not affect gypsy expression, in what other way could FOXO remedy the effect of gypsy expression? Does FOXO always induce the same set of genes, or could you do qPCR for different subsets of genes to see which ones are upregulated in the presence of gypsy expression (genes related to heat shock, antiviral response, stress response, etc)?

In Figure 1, since the focus of the figure is on TE expression, it could be helpful to put TEs that are up or down-regulated in a separate color.

In Figure 3, the panels should be labeled with lower case letters to match the other figures.

Use a consistent capitalization scheme when referring to GAL4.

---

## [Author Response]

The reviewers have discussed their reviews with one another, and the Reviewing Editor has drafted this to help you prepare a revised submission.Your revisions must address the major concerns of reviewer #2.1) You need to better explain your proposal as to how dFOXO actually works in the aging system you use. Is it a direct effect upon transposition? A discussion or clarification of this would be important. Reviewer #1 also asked if there might be cis-acting binding sites for FOXO and this may be relevant.

For this revision we performed two new experiments in which dFOXO was overexpressed in the context of the active gypsy element. We tested the paraquat resistance of the flies and found that dFOXO overexpression rescues the sensitivity to paraquat caused by gypsy expression. This result is consistent with our lab’s and many others work that show that dFOXO expression or activation is protective against oxidative stress.

We also measured the number of gypsy insertions in the dFOXO overexpression paradigm relative to the single copy Rp49 gene, as we did for the original experiment. We find dFOXO overexpression decreases the number of gypsy insertions. This role is new for the dFOXO pathway and we are working on what the downstream targets might be that lead to this effect. Taken together the results show dFOXO both rescues the oxidative stress induced by gypsy expression and decreases the insertions in the genome. We have included these results in an expanded Figure 8 and included new text in the manuscript describing them.

There are no known dFOXO cis-acting binding sites in either the UAS-gypsy construct or the gypsy LTR that would be affected by dFOXO overexpression.

2) You need to determine what the actual execution point during the life of the fly is when the effects of aging are manifest. Reviewer #2 points out that the experiments seem to not allow for the strong conclusions you have. Either modify with a clarification or address the point with a proviso.

We agree wholeheartedly with the reviewers; this is the real crux of the question. We would love to identify the window for the expression of the TE that affects lifespan. We have attempted this experiment using the geneswitch system (Osterwalder et al., PNAS 2001) for temporal control as well as the newer auxin-inducible gene expression system (AGES) (McClure et al., *eLife* 2022) for temporal control of UAS regulated genes. Unfortunately, in our hands, the small molecules, RU486 for geneswitch and auxin for AGES, affect the lifespan of the flies in our aging paradigm, even in the absence of GAL4. In Author response image 1 we show an example of the raw data for one of our replicates using the AGES experiments. The solid line are male flies of the genotype: *w1118*; UAS-Gypsy/+; PBac{tubP-TIR1-T2A-GAL80.AID}VK00040/+. Addition of auxin from eclosion (dotted line) or at 40 days (dashed line) shortens the lifespan of the animals in the absence of any GAL4. This makes it impossible to interpret the effect of gypsy expression. We have attempted this experiment multiple times with the same result. We are still working on ways to test this but as these longitudinal lifespan experiments take a very long time we wanted to report the principle finding. We believe the finding that increased transposon activity in somatic tissue affects lifespan and aging phenotypes is of interest to the field. It has been a long standing question whether TE expression itself is a burden that can shorten lifespan or if it is simply a byproduct of aging. Because we cannot define the window of effect at this point, we have included the requested proviso in the manuscript.

**Author response image 1. sa2fig1:** 

Reviewer #2 (Recommendations for the authors):Since FOXO overexpression does not affect gypsy expression, in what other way could FOXO remedy the effect of gypsy expression? Does FOXO always induce the same set of genes, or could you do qPCR for different subsets of genes to see which ones are upregulated in the presence of gypsy expression (genes related to heat shock, antiviral response, stress response, etc)?

Active dFOXO does not always induce the same set of genes. The response can be upstream signal specific. Currently, we do not know what the upstream signals that activate dFOXO are under increased transposon load. To get at what dFOXO signaling is doing downstream we investigated the oxidative stress response and gypsy integration. We have included new experiments in the figure 8 that show that dFOXO overexpression prevents the oxidative stress susceptibility seen in the wt animals. In addition, dFOXO overexpression decreases the number of insertion events.

In Figure 1, since the focus of the figure is on TE expression, it could be helpful to put TEs that are up or down-regulated in a separate color.

The mRNAs with expression changes are shown different colors (red and blue). We chose to make all transposons a third color to differentiate them from mRNAs. All of the transposons whose expression changes are statistically significant are identified by name in the figure.

In Figure 3, the panels should be labeled with lower case letters to match the other figures.

Thank you for pointing this out. We have edited the text to be consistent.

Use a consistent capitalization scheme when referring to GAL4.

Thank you for pointing this out. We have edited the text to be consistent.